# The clinical impact of comorbidities among patients with idiopathic pulmonary fibrosis undergoing anti-fibrotic treatment: A multicenter retrospective observational study

Ayako Aoki[1], Yu Hara[1]*, Hiroaki Fujii[1,2], Kota Murohashi[1], Ryo Nagasawa[3], Yoichi Tagami[1], Tatsuji Enomoto[4], Yutaka Matsumoto[5], Makoto Masuda[4,6], Keisuke Watanabe[1], Nobuyuki Horita[1], Nobuaki Kobayashi[1], Makoto Kudo[7], Takashi Ogura[3], Takeshi Kaneko[1]

1 Department of Pulmonology, Yokohama City University Graduate School of Medicine, Yokohama, Japan, 2 Department of Pulmonology, Yokohama Minami Kyousai Hospital, Yokohama, Japan, 3 Department of Respiratory Medicine, Kanagawa Cardiovascular and Respiratory Center, Yokohama, Japan, 4 Department of Respiratory Medicine, Ofuna Chuo Hospital, Kamakura, Japan, 5 Department of Respiratory Medicine, Yamato Municipal Hospital, Yamato, Japan, 6 Department of Respiratory Medicine, Fujisawa City Hospital, Fujisawa, Japan, 7 Respiratory Disease Center, Yokohama City University Medical Center, Yokohama, Japan

* yhara723@yokohama-cu.ac.jp

## Abstract

### Background

Among patients with idiopathic pulmonary fibrosis (IPF), few studies have investigated the clinical impact of anti-fibrotic treatment (AFT) with and without comorbidities. The aim of the study was to determine whether Charlson Comorbidity Index score (CCIS) can predict the efficacy of AFT in patients with IPF.

### Methods

We retrospectively assessed data extracted from the medical records of IPF patients who received anti-fibrotic agents between 2009 and 2019. The collected data included age, sex, CCIS, pulmonary function test, high-resolution computed tomography (HRCT) pattern, gender/age/physiology (GAP) score, and 3-year IPF-related events defined as the first acute exacerbation or death within 3 years after starting AFT.

### Results

We assessed 130 patients (median age, 74 years) who received nintedanib (n = 70) or pirfenidone (n = 60). Median duration of AFT was 425 days. Patients were categorized into high (≥ 3 points) and low (≤ 2 points) CCIS groups. There was no significant difference between the groups in terms of age, sex, duration of AFT, GAP score, or incidence of usual interstitial pneumonia pattern on HRCT except percentage predicted diffusion capacity of lung for carbon monoxide. Also, significant difference was not seen between the groups for 3-year IPF-

**Data Availability Statement:** We have attached the Raw Data to the Supplement information.

**Funding:** The author(s) received no specific funding for this work.

**Competing interests:** The authors have declared that no competing interests exist.

related events (P = 0.75). Especially, in the low CCIS group but not the high CCIS group, the longer duration of AFT had better disease outcome.

## Conclusion

In the present study, we could not show any relation between CCIS and IPF disease outcomes in patients undergoing AFT, though the longer duration of AFT might be beneficial for IPF outcomes among patients with low CCIS.

## Introduction

Idiopathic pulmonary fibrosis (IPF) occurs predominantly in older adults as a chronic, fibrosing idiopathic interstitial lung disease (ILD) having a chronic and progressive course characterized by aberrant accumulation of fibrotic tissue in the lung parenchyma and is associated with a steady worsening of respiratory symptoms and decline of pulmonary function [1–3]. The disease has a variable course; however, mortality is high with median survival of 2–5 years from the time of diagnosis, and the important causes of death are respiratory failure due to progression of IPF or acute exacerbation (AE) [4]. In addition to the progression itself, IPF has reported to be associated with numerous comorbidities, including coronary artery disease, gastroesophageal reflux disease, and lung cancer [5–7].

Anti-fibrotic treatment (AFT) with nintedanib or pirfenidone is approved worldwide for the treatment of IPF. These drugs have been associated with significant slowing of respiratory deterioration including AE in IPF and possibly also with prolonged survival [8, 9]. Further clinical trials have demonstrated that these drugs reduce the decline in lung function in patients with IPF, with consistent effects observed across the spectrum of baseline forced vital capacity (FVC) that was studied (FVC > 50% predicted), and across subgroups by age, race, sex, and concomitant medication use [10]. Despite clinical efficacy was shown in the several research, for physicians, it is difficult to manage the adverse reactions including liver disfunction, gastrointestinal adverse events such as diarrhoea, nausea, and appetite loss.

Although the mechanism of IPF progression is not well known, a previous study has demonstrated that these comorbidities can have a significant influence on the symptoms and quality of life of patients with IPF, particularly when multiple comorbidities are present [11]. Furthermore, in the previous studies, the progression of comorbidities may be pathophysiologically linked to the progression of IPF itself, though their prognostic impact and mechanism is not fully understood [12–15]. The difficulty of management for pirfenidone and nintedanib, the high incidence of older adults, and the various comorbidities seems to affect the tolerability and clinical efficacy of AFT, however, few studies have investigated the clinical impact of AFT in patients with and without comorbidities or evaluated its importance in terms of the long-term prognosis of IPF [16].

The aim of this multicenter retrospective observational study was to evaluate whether Charlson Comorbidity Index score (CCIS) could affect the disease outcomes undergoing AFT including nintedanib and pirfenidone in patients with IPF [17].

## Methods

### Study location and enrolled patients

This retrospective, observational study was performed using data from consecutive patients diagnosed with IPF who underwent AFT at Yokohama City University Hospital, Yokohama

City University Medical Center, Yokohama Minami Kyousai Hospital, Kanagawa Cardiovascular and Respiratory Center, Ofuna Chuo Hospital, Yamato Municipal Hospital, and Fujisawa City Hospital, Japan, between 2009 and 2019. The diagnosis of IPF by individual physicians was based on the criteria proposed by the official guideline of the American Thoracic Society/European Respiratory Society/Japanese Respiratory Society/Latin American Thoracic Society [2]. An AE was defined as acute respiratory deterioration requiring steroid pulse therapy adding on AFT including clinical worsening of dyspnea, hypoxemia, or the worsening or severe impairment of gas exchange characterized by new bilateral ground-glass opacification/consolidation superimposed on a background pattern consistent with IPF pattern not fully explained by cardiac failure or fluid overload [4].

## Data collection

The following were collected from a review of the patients' medical records: age; sex; smoking history; CCIS; pulmonary function test; high-resolution computed tomography (HRCT) pattern including usual interstitial pneumonia (UIP), probable UIP, and indeterminate UIP patterns; gender/age/physiology (GAP) score; and anti-fibrotic agent (nintedanib or pirfenidone) at the time of starting AFT [2, 18]. Also, we evaluated the duration of AFT and disease outcome. The disease outcome was 3-year IPF-related events, defined as IPF-related mortality due to respiratory failure or the first AE within 3 years after initiating AFT. The CCIS is an established and widely used tool to measure comorbidity burden with 19 different medical conditions [17]. It is scored based on the number and severity of comorbidities, with higher CCI indicating greater comorbidity burden and severity. It was developed to assess the risk of death from comorbidities and has been widely applied as a prognostic indicator for patients with colorectal cancer, advanced non-small cell lung carcinoma, and acute myocardial infarction [19–21].

## Statistical analysis

No statistical sample size calculations were conducted because of the retrospective nature. Groups were compared using chi-square test and Wilcoxon rank-sum test (non-parametric test) because a normal distribution cannot be assumed for data obtained with a small number of samples. Univariate and multivariate analyses were performed to identify primary predictors of 3-year IPF-related events, including cause-specific mortality and first AE. When comparing 3-year IPF-related events among the low ($\leq$ 2 points) and high CCIS ($\geq$ 3 points) groups, survival curves were generated using the Kaplan–Meier method and compared using the log-rank test. The enrolled patients were divided into the low CCIS group ($\leq$ 2 points) and the high CCIS group ($\geq$ 3 points), because the average and median values of CCIS from our data were 2.2 and 2 points, respectively. All statistical analyses were performed using JMP12 (SAS Institute, Cary, NC). Values are expressed as the median (25th–75th percentiles) or the number (%). Values of $P < 0.05$ were considered to indicate statistical significance.

## Ethics approval

This study followed the guidelines of the Declaration of Helsinki and was approved by the institutional review board (IRB) at Yokohama City University Hospital (approval No. B190300032). Due to the retrospective nature, the enrolled patients provide their verbal informed consent to participate in this study. The description of the study was provided to these patients on an opt-out basis via the Yokohama City University website (https://www.yokohama-cu.ac.jp/amedrc/ethics/ethical/fuzoku_optout.html). In addition, IRB-approved procedures for the description of this study have been obtained.

## Results

### Patient characteristics

A flowchart of the participant selection process is shown in Fig 1. Among 196 patients identified with IPF, excluded were patients without pulmonary function data, those who had started AFT after AE, and those without medical information regarding comorbidities. Of the final total of 130 patients, 70 were treated with nintedanib and 60 with pirfenidone. S1 Table showed no significant difference of baseline data including age, sex, CCIS, pulmonary function, severity, the timing of starting AFT between nintedanib and pirfenidone groups, though the different launch dates of pirfenidone (in 2008) and nintedanib (in 2015) in Japan may have contributed to be the selection bias exists for both patient and physician as shown in the S2 Fig. The patients were divided into two groups according to CCIS: low CCIS group ($\leq$ 2 points, N = 88) and high CCIS group ($\geq$ 3 points; N = 42). Table 1 lists the clinical characteristics of the 130 patients with IPF (median age, 74 years; male, 78%; median CCIS, 2 points; proportion with HRCT pattern of UIP or probable UIP, 79%; median GAP score, 3 points). There was no significant difference between the high and low CCIS groups for any of these parameters except percentage predicted diffusion capacity of lung for carbon monoxide (%$D_{Lco}$). Median treatment duration was 358 days for nintedanib and 410 days for pirfenidone. The duration of nintedanib treatment tended to be longer in the low CCIS group than in the high CCIS group (437 days vs. 255 days (P = 0.14)). As shown in Table 2, GAP score and duration

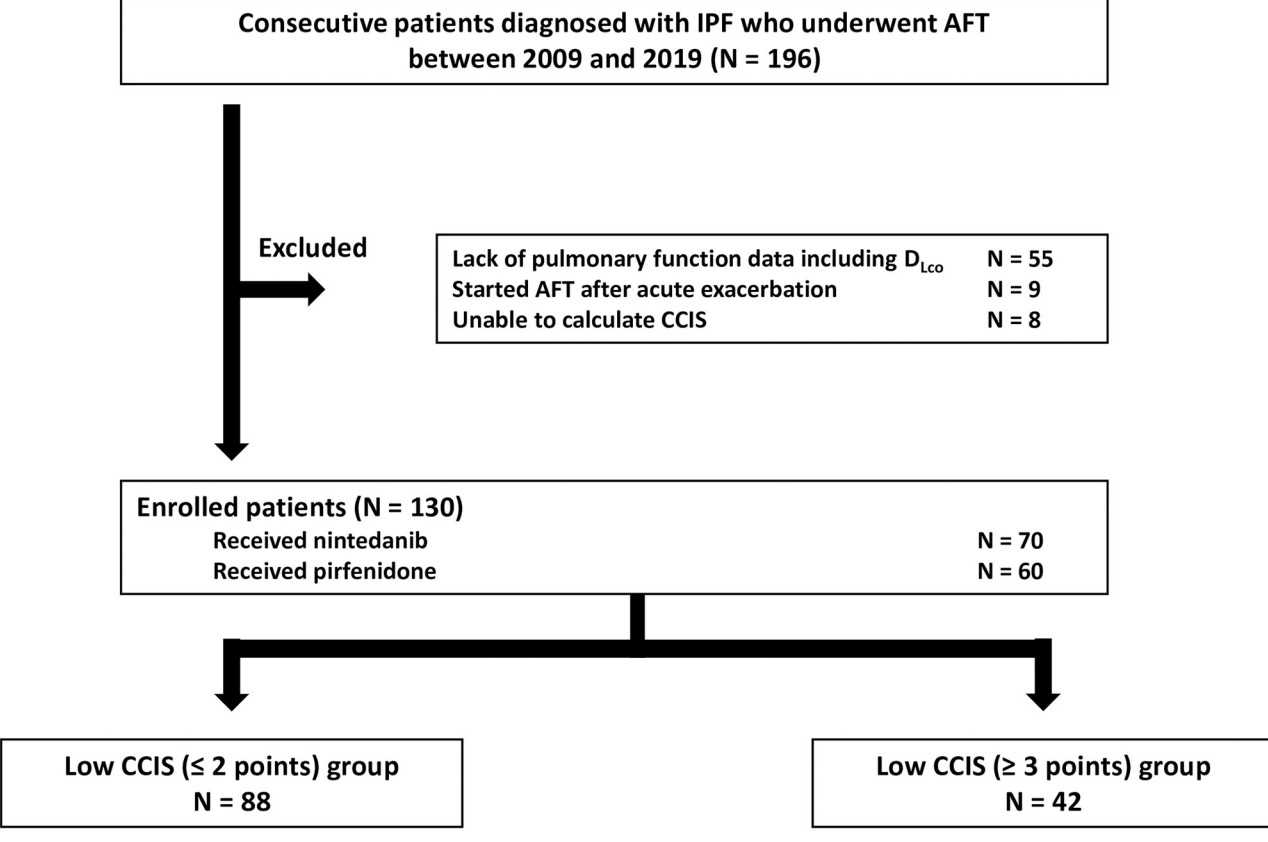

**Fig 1. Flowchart of the participants selection process. Abbreviations:** AFT, anti-fibrotic treatment; CCIS, Charlson Comorbidity Index Score; IPF, idiopathic pulmonary fibrosis.

**Table 1. Patients' characteristics.**

| Variable | Low CCIS (N = 88) | High CCIS (N = 42) | Overall (N = 130) | P value (low vs. high) |
|---|---|---|---|---|
| **Age** | 74 (69–78) | 74 (69–77) | 74 (69–77) | 0.60 |
| **Male, N (%)** | 69 (78) | 32 (76) | 101 (78) | 0.78 |
| **Never smoker, N (%)** | 27 (31) | 8 (20) | 35 (27) | 0.18 |
| **CCIS** | 1 (1–2) | 4 (3–5) | 2 (1–3) | < 0.01 |
| Cardiovascular disease, N (%) | 9 (10) | 13 (31) | 22 (17) | < 0.01 |
| Malignancy, N (%) | 0 (0) | 22 (52) | 22 (17) | < 0.01 |
| **%FVC** | 78 (67–91) | 75 (64–93) | 77 (65–91) | 0.92 |
| **%$D_{Lco}$** | 63 (47–75) | 51 (38–68) | 61 (44–71) | 0.02 |
| **HRCT** | | | | |
| UIP pattern, N (%) | 59 (67) | 32 (76) | 91 (70) | 0.28 |
| Probable UIP pattern, N (%) | 10 (11) | 2 (5) | 12 (9) | 0.22 |
| **GAP score** | 3 (3–4) | 4 (3–4) | 3 (3–4) | 0.11 |
| **Anti-fibrotic agent** | | | | 0.60 |
| Nintedanib, N (%) | 46 (66) | 24 (34) | 70 (100) | |
| Pirfenidone, N (%) | 42 (70) | 18 (30) | 60 (100) | |
| From diagnosis to starting AFT, days | 298 (73–748) | 221 (26–451) | 260 (63–652) | 0.22 |
| Duration of AFT, days | 424 (128–867) | 349 (52–537) | 393 (91–695) | 0.05 |
| Adverse reaction | | | | 0.89 |
| GI symptoms, N (%) | 14 (15) | 9 (21) | 23 (18) | |
| Liver disfunction, N (%) | 6 (7) | 2 (5) | 8 (6) | |
| Others, N (%) | 5 (6) | 5 (12) | 10 (8) | |
| Dose down or stop AFT (N/P) | 25 (28) / 17 (19) | 13 (31) / 14 (33) | 38 (29) / 31 (24) | 0.07 |
| **Outcomes** | | | | |
| Follow-up duration, days | 476 (245–933) | 489 (181–748) | 476 (224–890) | 0.41 |
| Acute exacerbation, N (%) | 26 (30) | 13 (31) | 39 (30) | 0.90 |
| IPF related events, N (%) | 41 (47) | 19 (45) | 60 (46) | 0.89 |

**Footnote:** Data are presented as the median (25th–75th percentiles) or the number (%).

**Abbreviations:** AFT, anti-fibrotic treatment; CCIS, Charlson Comorbidity Index Score; GAP, gender / age / physiology; GI, gastrointestinal; HRCT, high resolution computed tomography; IPF, idiopathic pulmonary fibrosis; N, nintedanib; P, pirfenidone; %$D_{Lco}$, percentage predicted diffusion capacity of lung for carbon monoxide, %FVC, percentage predicted forced vital capacity; UIP, usual interstitial pneumonia.

**Table 2. Univariate and multivariate analysis of primary predictors of 3-year IPF related events.**

| Variable | Hazard ratio | 95% CI | P | Hazard ratio | 95% CI | P |
|---|---|---|---|---|---|---|
| **Age** | 0.954 | 0.911–0.999 | 0.04 | 0.975 | 0.945–1.008 | 0.13 |
| **Sex (male vs. female)** | 0.976 | 0.442–2.152 | 0.95 | | | |
| **CCIS** | 0.926 | 0.768–1.102 | 0.40 | | | |
| **%FVC** | 1.002 | 0.985–1.022 | 0.79 | | | |
| **%$D_{Lco}$** | 1.010 | 0.989–1.030 | 0.33 | | | |
| **GAP score** | 1.796 | 1.188–2.758 | 0.01 | 1.524 | 1.250–1.849 | < 0.01 |
| **UIP pattern** | 1.357 | 0.724–2.542 | 0.33 | | | |
| **Anti-fibrotic agents (N vs. P)** | 1.043 | 0.615–1.770 | 0.87 | | | |
| **Duration of AFT** | 0.999 | 0.998–0.999 | < 0.01 | 0.999 | 0.998–1.000 | < 0.01 |

**Abbreviations:** AFT, anti-fibrotic treatment; CCIS, Charlson Comorbidity Index Score; CI, confidence interval; GAP, gender / age / physiology; IPF, idiopathic pulmonary fibrosis; N, nintedanib; P, pirfenidone; %$D_{Lco}$, percentage predicted diffusion capacity of lung for carbon monoxide, %FVC, percentage predicted forced vital capacity; UIP, usual interstitial pneumonia.

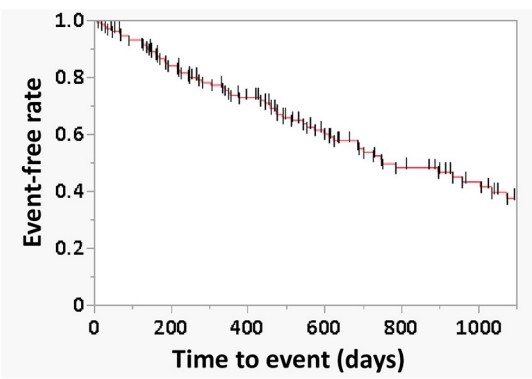

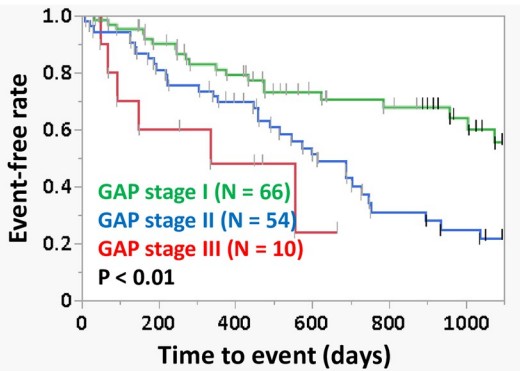

**Fig 2. Survival curves of 130 patients (A, All patients: B, according to GAP stage).** Comparison of the curves by log-rank test showed significant difference among the GAP stage groups (P < 0.01). **Abbreviation:** GAP, gender/age/physiology.

of AFT proved to be the significant predictors among 130 patients. Fig 2 shows Kaplan–Meier survival curves for 3-year IPF-related events according to GAP stage. Comparison by log-rank test revealed significant difference among the groups (P < 0.01). In addition, there was no significant difference in CCIS according to GAP stage (CCIS of GAP stage I, 2 points; II, 2 points; III, 2.5 points (P = 0.32)).

## Comparison of 3-year IPF-related events between CCIS groups

Fig 3A shows Kaplan–Meier survival curves for 3-year IPF-related events for the high CCIS and low CCIS groups among 130 patients. Comparison by log-rank test showed no significant difference between the groups (P = 0.75). Similar trends were also observed in 169 patients with evaluable CCIS and 3-year IPF-related events (S1 Fig). Survival curves were also compared between the high and low CCIS groups for each anti-fibrotic agent (Fig 3B and 3C). In the patients treated with nintedanib, log-rank test showed no significant difference in 3-year IPF-related events according to CCIS group (high CCIS group, N = 24; low CCIS group, N = 46; P = 0.43) (Fig 3B). A similar tendency was seen in the patients treated with pirfenidone (high CCIS group, N = 18; low CCIS group, N = 42; P = 0.62) (Fig 3C).

## Univariate and multivariate analysis of primary predictors of 3-year IPF-related events

Univariate and multivariate analysis were performed to determine the primary predictors of 3-year IPF-related events, using the following parameters: age, sex, CCIS, %$D_{Lco}$, %FVC, GAP score, HRCT UIP pattern, anti-fibrotic agent (nintedanib vs. pirfenidone), and duration of AFT (Table 2). Multivariate analysis showed GAP score and duration of AFT as the significant predictors among all patients. As shown in Table 3, we also performed univariate and multivariate analysis according CCIS. In the high CCIS group, univariate analysis showed that only GAP score and %$D_{Lco}$ were significant predictors of 3-year IPF-related events. In the low CCIS group, age and duration of AFT were significant predictors. Multivariate analysis showed GAP score in the high CCIS group and duration of AFT in the low CCIS group as the significant predictors. Especially, in the low CCIS group, comparison of survival curves between patients with duration of AFT < 1 year and ≥ 1 year showed significant differences

## A. All patients

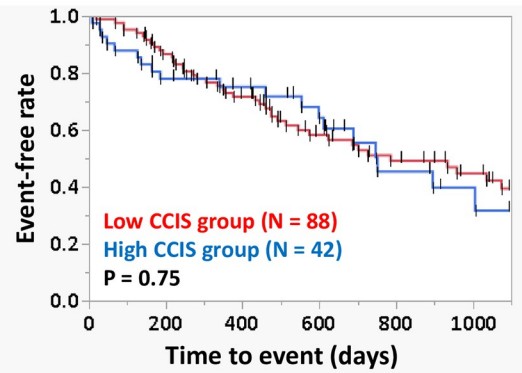

## B. Nintedanib group

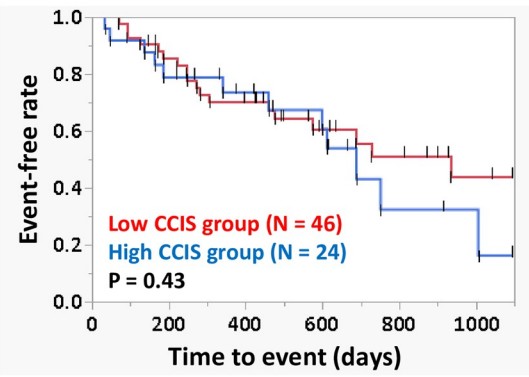

## C. Pirfenidone group

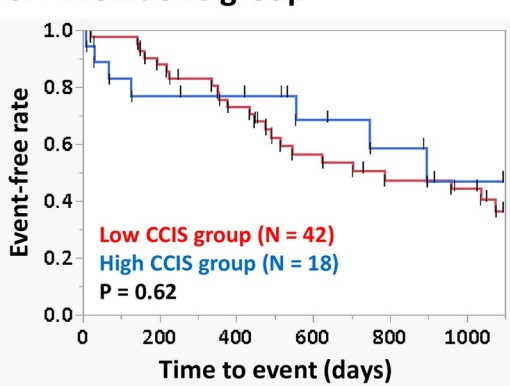

**Fig 3. Survival curves for 3-year IPF-related events according to CCIS and anti-fibrotic agent among 130 patients.** Comparison of the curves by log-rank test in 130 patients according to CCIS group (A), and according to CCIS group in patients treated with nintedanib (B) and pirfenidone (C) showed no significant difference (P = 0.75, P = 0.43, and P = 0.62, respectively). **Abbreviations:** CCIS, Charlson Comorbidity Index Score; IPF, idiopathic pulmonary fibrosis.

(P < 0.01), though there was no significant difference of patients' characteristics between these groups except %$D_{Lco}$ (Table 4, Fig 4A and 4B).

## Discussion

Nintedanib and pirfenidone are currently approved as AFT agents for IPF based on the results of large scale phase III clinical trials [8, 9]. There is also accumulating evidence of clinical experience with AFT in the real world, in a patient population that is generally older, with comorbidities, and who may not meet the stringent criteria for clinical trial inclusion [22–25]. Here we evaluated whether CCIS could affect the disease outcomes undergoing AFT including nintedanib and pirfenidone in patients with IPF in the present study.

Comorbidities appear to be important predictors in determining the prognosis of IPF. Kakugawa et al. reported that cardiovascular disease, congestive heart failure, and lung cancer were significant predictors of AE onset and of prognosis [26]. Another study has suggested significant negative impacts of arteriosclerosis, other cardiovascular diseases (mainly valvular heart disease, cardiac arrhythmias, and dilated cardiomyopathy), and lung cancer comorbidities on survival in patients with IPF [8]. In our retrospective studies, we demonstrated that CCIS was an important prognostic indicator among patients with ILD, in both AE and stable

**Table 3. Univariate and multivariate analysis of primary predictors of 3-year IPF related events in the patients with high CCIS and low CCIS.**

| High CCIS | Univariate | | | Multivariate | | |
|---|---|---|---|---|---|---|
| Variable | Hazard ratio | 95% CI | P | Hazard ratio | 95% CI | P |
| Age | 0.931 | 0.825–1.057 | 0.26 | | | |
| Sex (male vs. female) | 0.519 | 0.101–2.672 | 0.44 | | | |
| %FVC | 1.034 | 0.997–1.071 | 0.07 | | | |
| %$D_{Lco}$ | 1.055 | 1.008–1.109 | 0.02 | 1.020 | 0.987–1.051 | 0.231 |
| GAP score | 5.086 | 1.694–18.25 | < 0.01 | 2.119 | 1.252–3.807 | < 0.01 |
| UIP pattern | 1.111 | 0.311–0.397 | 0.87 | | | |
| Anti-fibrotic agents (N vs. P) | 2.810 | 0.786–10.053 | 0.10 | | | |
| Duration of AFT | 0.999 | 0.997–1.000 | 0.11 | | | |
| Low CCIS | Univariate | | | Multivariate | | |
| Variable | Hazard ratio | 95% CI | P | Hazard ratio | 95% CI | P |
| Age | 0.936 | 0.885–0.990 | 0.02 | 0.973 | 0.938–1.009 | 0.14 |
| Sex | 1.031 | 0.414–2.566 | 0.95 | | | |
| %FVC | 0.990 | 0.970–1.010 | 0.34 | | | |
| %$D_{Lco}$ | 1.002 | 0.975–1.031 | 0.86 | | | |
| GAP score | 1.523 | 0.942–2.462 | 0.08 | | | |
| UIP pattern | 1.302 | 0.629–2.696 | 0.48 | | | |
| Anti-fibrotic agents (N vs. P) | 0.888 | 0.459–1.718 | 0.72 | | | |
| Duration of AFT | 0.998 | 0.997–0.999 | < 0.01 | 0.998 | 0.997–0.999 | < 0.01 |

**Abbreviations:** AFT, anti-fibrotic treatment; CCIS, Charlson Comorbidity Index Score; CI, confidence interval; GAP, gender / age / physiology; IPF, idiopathic pulmonary fibrosis; N, nintedanib; P, pirfenidone; %$D_{Lco}$, percentage predicted diffusion capacity of lung for carbon monoxide, %FVC, percentage predicted forced vital capacity; UIP, usual interstitial pneumonia.

**Table 4. Patients' characteristics between the duration of AFT < 1 yr and ≥ 1 yr groups.**

| Variable | Low CCIS group | | | High CCIS group | | |
|---|---|---|---|---|---|---|
| | < 1 yr (N = 41) | ≥ 1 yr (N = 47) | P value (< 1 vs. ≥ 1 yr) | < 1 yr (N = 22) | ≥ 1 yr (N = 20) | P value (< 1 vs. ≥ 1 yr) |
| Age | 74 (71–80) | 74 (66–77) | 0.08 | 74 (69–77) | 74 (68–78) | 0.81 |
| Male, N (%) | 34 (83) | 35 (74) | 0.34 | 16 (72) | 16 (80) | 0.58 |
| %FVC | 77 (64–89) | 78 (69–93) | 0.59 | 75 (64–88) | 75 (69–101) | 0.52 |
| %$D_{Lco}$ | 70 (52–88) | 58 (43–70) | < 0.01 | 54 (37–70) | 51 (40–60) | 0.98 |
| HRCT | | | | | | |
| UIP pattern, N (%) | 26 (63) | 33 (70) | 0.50 | 16 (72) | 16 (80) | 0.58 |
| GAP score | 3 (3–4) | 2 (3–4) | 0.55 | 4 (3–4) | 4 (3–5) | 0.94 |
| Anti-fibrotic agent | | | 0.85 | | | 0.37 |
| Nintedanib, N (%) | 21 (46) | 25 (54) | | 14 (58) | 10 (42) | |
| Pirfenidone, N (%) | 20 (48) | 22 (52) | | 8 (44) | 10 (56) | |
| From diagnosis to starting AFT, days | 323 (67–753) | 280 (74–806) | 0.92 | 230 (39–392) | 211 (16–881) | 0.90 |
| Outcomes | | | | | | |
| Follow-up duration, days | 247 (146–486) | 624 (471–1075) | < 0.01 | 299 (111–632) | 608 (460–854) | 0.04 |
| Acute exacerbation, N (%) | 13 (32) | 14 (30) | 0.85 | 8 (36) | 5 (25) | 0.43 |
| IPF related events, N (%) | 21 (51) | 20 (43) | 0.42 | 10 (45) | 9 (45) | 0.98 |

**Footnote:** Data are presented as the median (25th–75th percentiles) or the number (%).

**Abbreviations:** AFT, anti-fibrotic treatment; CCIS, Charlson Comorbidity Index Score; GAP, gender / age / physiology; HRCT, high resolution computed tomography; %$D_{Lco}$, percentage predicted diffusion capacity of lung for carbon monoxide, %FVC, percentage predicted forced vital capacity; UIP, usual interstitial pneumonia.

## A. Low CCIS group

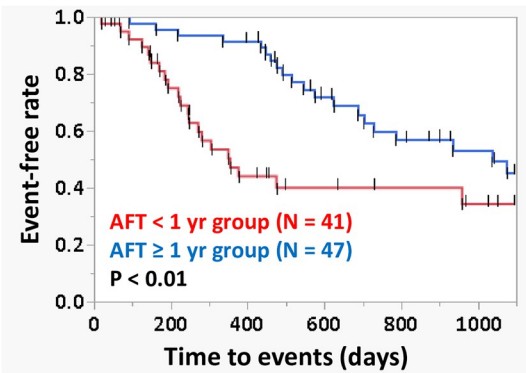

## B. High CCIS group

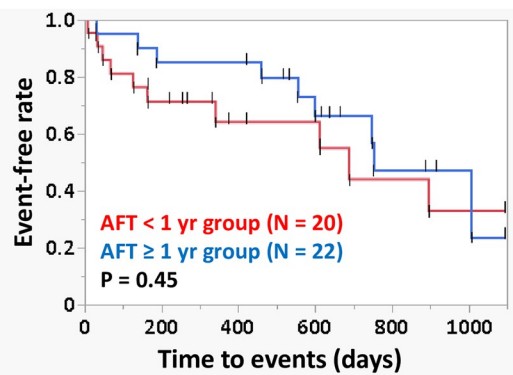

**Fig 4. Survival curves of patients with the duration of AFT < 1 yr and ≥ 1 yr groups.** Comparison of the curves by log-rank test showed significant difference in the low CCIS group (P < 0.01). **Abbreviations:** AFT, anti-fibrotic treatment; CCIS, Charlson Comorbidity Index Score.

conditions [27, 28]. Also, in the recent research the progression of comorbidities reported to be pathophysiologically linked to the progression of IPF itself, though their prognostic impact and mechanism is not fully understood [12–15]. Previous studies have revealed a high incidence of lung cancer in IPF (7% to 20%), though the true cumulative incidence of lung cancer after the diagnosis of IPF and its predictive factors at the initial diagnosis of IPF remain unknown. Various mechanisms such as endoplasmic reticulum stress, alterations of growth factors expression, oxidative stress, and large genetic and epigenetic variations, myofibroblast/mesenchymal transition, myofibroblast activation and proliferation can contributed to predispose the patient to develop IPF and lung cancer [12]. MicroRNAs (miRNAs) regulate gene expression at the post-transcriptional level contributing to all major cellular processes, including oxidative stress and cell death. Several miRNAs have been reported to cross-talk with oxidative stress in both cardiac and pulmonary systems [13]. Fibrogenic mediators such as transforming growth factor-β promotes fibroblast migration, proliferation, and activation in the heart and lungs [14]. Mechanisms contributing to the development of pulmonary hypertension in patients with IPF are complex including hypoxia causing smooth muscle hypertrophy and collagen deposition in pulmonary arteries, the destruction and obstruction of pulmonary vasculature by the progression of pulmonary fibrosis, and vascular remodeling contributed by fibroblast growth factor and platelet-derived growth factor [15]. From these, the progression of IPF seems to cross-talk with other comorbidities, suggesting that comorbidities may have an impact on the efficacy of AFT and high CCIS not only indicates an increased risk of death from comorbidities but also poorer prognosis for IPF itself. However, the present study revealed that among IPF populations undergoing nintedanib or pirfenidone whose prognosis had worsened according to GAP stage, CCIS did not have significant impact on 3-year IPF-related events. There are several reasons why CCIS was found not to be a prognostic indicator for IPF. First, there was no significant difference in CCIS according to GAP stage and the CCIS of enrolled patients was relatively low. Second, whereas Yagyu et al. reported that CCIS was useful for predicting long-term prognosis in ILD patients with relatively preserved pulmonary function (%FVC, 94%; %$D_{Lco}$, 92%), in the present patients, the severity of ILD and impairment of pulmonary function (%FVC, 78%; %$D_{Lco}$, 63%) was estimated to be greater than the severity of extrapulmonary comorbidities [28]. Actually, in a previous cohort study evaluating efficacy of nintedanib under several comorbidities, nintedanib was shown to reduce the rate of decline in FVC over 52 weeks versus placebo in patients with CCIS ≤ 3 (difference:

106.4 ml/year) and CCIS > 3 (difference: 129.5 ml/year) (P = 0.57 for treatment-by-time-by-subgroup interaction). These findings suggest that the effectiveness of nintedanib for reducing the rate of FVC decline is consistent across subgroups based on comorbidity burden [12].

Long-term AFT may have an impact on IPF survival. In their retrospective study of 104 patients with IPF who underwent treatment with nintedanib, Kato et al. reported that the median survival time in patients with short-term AFT (< 12 months) was significantly shorter than that in patients with long-term AFT (≥ 12 months) [29]. Consistent with this observation, in the low CCIS group log-rank tests showed significant difference in the Kaplan–Meier survival curves between patients with duration of AFT < 1 year and ≥ 1 year. As shown in Table 4, in the low CCIS group, the patients with duration of AFT ≥ 1 year group (lower % $D_{Lco}$ populations than those with duration of AFT < 1 year) had favorable disease outcome, though there was no significant difference in the duration of AFT < 1 year and ≥ 1 year in the GAP score and the time from diagnosis to starting AFT. From these, AFT may be expected to prolong prognosis even in the patients with relatively advanced IPF under appropriate management of comorbidities.

The present study demonstrated that CCIS could not influence the disease outcomes of IPF patients undergoing AFT. However, this study has several limitations. First, this study is a retrospective cohort study with small number IPF patients and no statistical sample size calculations were also conducted because of the retrospective nature. then the reproducibility of the findings of this study needs to be confirmed through validation cohorts which increased the number of cases in the future. Second, the timing of initiating AFT and whether to use nintedanib or pirfenidone depends on the physicians' decision. Actually, from Table 1, %$D_{Lco}$ was preserved in the low CCIS group compared to the high CCIS group, suggesting that AFT was introduced in relatively mild IPF patients in the low CCIS group. In general, for both patient and physician, it is difficult to use anti-fibrotic agents due to their high cost and the difficulty of management of the adverse reaction. From these, it seems that selection bias exists. Third, the diagnosis of IPF depended on individual physicians based on the criteria proposed by official guideline despite the majority of cases having a typical UIP pattern on HRCT. Although this study is real world data, a multi-disciplinary discussion may be necessary for a rigorous diagnosis of IPF. Fourth, the comorbidities such as lung cancer, cardiovascular disease, and pulmonary hypertension is very important treatable traits. As the future research, we should evaluate whether early introduction of AFT and rigorous comorbidity management improve the long-term prognosis of IPF.

## Conclusions

In the present study, we could not show any relation between CCIS and IPF disease outcomes in patients undergoing AFT, though the longer duration of AFT might be beneficial for IPF outcomes among patients with low CCIS.

## Supporting information

**S1 Data.**
(XLSX)

**S1 Fig. The changes in the number of prescriptions for antifibrotic agents over time.**
(PPTX)

**S2 Fig. Survival curves for 3-year IPF-related events according to CCIS and anti-fibrotic agent among 169 patients.** Comparison of the curves by log-rank test in 196 patients according to CCIS group (A), and according to CCIS group in patients treated with nintedanib (B)

and pirfenidone (C) showed no significant difference (P = 0.75, P = 0.43, and P = 0.62, respectively). Abbreviations: CCIS, Charlson Comorbidity Index Score; IPF, idiopathic pulmonary fibrosis.
(PPTX)

**S3 Fig. Survival curves for 3-year IPF-related mortality according to CCIS and anti-fibrotic agent among 130 patients.** Comparison of the curves by log-rank test in 130 patients according to CCIS group (A), and according to CCIS group in patients treated with nintedanib (B) and pirfenidone (C) showed no significant difference (P = 0.30, P = 0.21, and P = 0.91, respectively). Abbreviations: CCIS, Charlson Comorbidity Index Score; IPF, idiopathic pulmonary fibrosis.
(PPTX)

**S1 Table. Patients' baseline data for nintedanib or pirfenidone groups.**
(PPTX)

## Author Contributions

**Conceptualization:** Ayako Aoki, Yu Hara, Hiroaki Fujii, Kota Murohashi, Yoichi Tagami, Tatsuji Enomoto, Yutaka Matsumoto, Makoto Masuda, Keisuke Watanabe, Nobuyuki Horita, Nobuaki Kobayashi, Makoto Kudo, Takashi Ogura, Takeshi Kaneko.

**Data curation:** Ayako Aoki, Yu Hara, Hiroaki Fujii, Kota Murohashi, Ryo Nagasawa, Yoichi Tagami, Tatsuji Enomoto, Yutaka Matsumoto, Makoto Masuda, Keisuke Watanabe, Nobuyuki Horita, Nobuaki Kobayashi, Makoto Kudo, Takashi Ogura, Takeshi Kaneko.

**Formal analysis:** Ayako Aoki, Yu Hara, Kota Murohashi, Ryo Nagasawa, Yoichi Tagami, Tatsuji Enomoto, Yutaka Matsumoto, Makoto Masuda, Keisuke Watanabe, Nobuyuki Horita, Nobuaki Kobayashi, Makoto Kudo, Takashi Ogura, Takeshi Kaneko.

**Funding acquisition:** Ayako Aoki, Yu Hara, Hiroaki Fujii, Kota Murohashi, Ryo Nagasawa, Yoichi Tagami, Keisuke Watanabe, Nobuyuki Horita, Nobuaki Kobayashi, Takeshi Kaneko.

**Investigation:** Ayako Aoki, Yu Hara, Hiroaki Fujii, Kota Murohashi, Ryo Nagasawa, Yoichi Tagami, Tatsuji Enomoto, Yutaka Matsumoto, Makoto Masuda, Keisuke Watanabe, Nobuyuki Horita, Nobuaki Kobayashi, Makoto Kudo, Takashi Ogura, Takeshi Kaneko.

**Methodology:** Ayako Aoki, Yu Hara, Yoichi Tagami, Tatsuji Enomoto, Yutaka Matsumoto, Makoto Masuda, Keisuke Watanabe, Nobuyuki Horita, Nobuaki Kobayashi, Takashi Ogura, Takeshi Kaneko.

**Project administration:** Ayako Aoki, Yu Hara, Hiroaki Fujii, Nobuyuki Horita, Takeshi Kaneko.

**Resources:** Ayako Aoki, Yu Hara, Kota Murohashi, Ryo Nagasawa, Yoichi Tagami, Yutaka Matsumoto, Makoto Masuda, Makoto Kudo, Takashi Ogura, Takeshi Kaneko.

**Software:** Yu Hara.

**Supervision:** Yu Hara, Yoichi Tagami, Takeshi Kaneko.

**Validation:** Ayako Aoki, Yu Hara, Hiroaki Fujii, Kota Murohashi, Ryo Nagasawa, Yoichi Tagami, Nobuyuki Horita, Nobuaki Kobayashi.

**Visualization:** Yu Hara.

**Writing – original draft:** Ayako Aoki, Yu Hara.

**Writing – review & editing:** Yu Hara, Hiroaki Fujii, Kota Murohashi, Ryo Nagasawa, Yoichi Tagami, Tatsuji Enomoto, Yutaka Matsumoto, Makoto Masuda, Keisuke Watanabe, Nobuyuki Horita, Nobuaki Kobayashi, Makoto Kudo, Takashi Ogura, Takeshi Kaneko.

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
