## [Decision Letter · Decision Letter 0]

3 Feb 2023

PONE-D-22-31881

Charlson Comorbidity Index score does not predict the duration and efficacy of antifibrotic treatment for idiopathic pulmonary fibrosis: A multicenter retrospective observational study

PLOS ONE

Dear Dr. Hara,

Thank you for submitting your manuscript to PLOS ONE. After careful consideration, we feel that it has merit but does not fully meet PLOS ONE’s publication criteria as it currently stands. Therefore, we invite you to submit a revised version of the manuscript that addresses the points raised during the review process.

We look forward to receiving your revised manuscript.

Kind regards,

Latika Gupta

Academic Editor

PLOS ONE

Journal Requirements:

a) Did participants provide their written or verbal informed consent to participate in this study?

6. Please include your tables as part of your main manuscript and remove the individual files. Please note that supplementary tables (should remain/ be uploaded) as separate "supporting information" files

Additional Editor Comments:

Please revise based on reviewer comments as appended.

Reviewers' comments:

Reviewer's Responses to Questions

**Comments to the Author**

1. Is the manuscript technically sound, and do the data support the conclusions?

Reviewer #1: Partly

Reviewer #2: No

2. Has the statistical analysis been performed appropriately and rigorously? 

Reviewer #1: Yes

Reviewer #2: Yes

3. Have the authors made all data underlying the findings in their manuscript fully available?

Reviewer #1: Yes

Reviewer #2: Yes

4. Is the manuscript presented in an intelligible fashion and written in standard English?

Reviewer #1: Yes

Reviewer #2: Yes

5. Review Comments to the Author

Reviewer #1: Dear Editor,

Thank you for the opportunity to review this paper entitled “Charlson Comorbidity Index score does not predict the duration and efficacy of antifibrotic treatment for idiopathic pulmonary fibrosis: A multicenter retrospective observational study”. I read with interest this paper, which shows the impact of comorbidities evaluated with CCIS in IPF patients treated with AFTs. The results showed that CCIS is not able to differentiate patients between those with good and bad prognoses. However, the authors do not explain why they have selected a CCSI value of 3 to divide the two groups and what types of AEs were considered for outcome definition.

In my opinion, this study has some limitations that have to be improved.

Methods

- The authors should explain why a threshold of 3 was chosen for patient selection in low and high CCSI groups. It is possible that and higher value of CCSI could impact on the survival or duration of treatment. I suggest performing a cox regression analysis using CCSI as continuous variable.

- What AEs were collected and considered as disease outcomes for the purpose of this study? This data is not explained in the methods.

Results

- Add the list of comorbidities and a comparison between the two group

- Add a list of AEs recorded during follow-up and compare the two group

Discussion

- Authors showed that in low CCSI group, but not in high CCIS group, AFT treatment <1yr impacts on survival. This result deserves to be further explored in the discussion

Reviewer #2: This is a retrospective review assessing whether comorbidities predict efficacy of antifibrotics using the Charlson Comorbidity Index score.

This is a multi-centre study over a 10 year period.

Only 130 patients are included in this 10 year period – approximately 13 patients per year which is a very low number of patients for a multicentre study – 7 described here. The methods describe that consecutive patients were recruited during this time period. 196 were identified which is approximately 16 IPF patients in 7 centres per month.

The study showed no difference in outcome based on whether patients were stratified according to low or high CCIS score.

Strengths of paper:

This paper is original as I’m not aware that any papers have published on this before. Despite this I’m unsure as to the rationale for why comorbidities would impact on the efficiency of antifibrotic therapy and the authors have failed to explain this rationale in the introduction (although its later described in discussion)

Clearly and succinctly written with good use of English language

Figures and tables are clear

Weaknesses of paper.

Small numbers in 7 centres in Japan limit the generalisability of the results

Limited information in methods on how diagnosis is made and the impact that comorbidities may have on prescribing the antifibrotics or decision making on which antifibrotic is prescribed which is clearly evident in the results. The process isnt standardised and left to the physicians which introduces a lot of bias

Retrospective review brings in a lot of bias re lead time bias ; diagnostic bias which limits any conclusions from the study and this needs to be tempered in the discussion

A significant number of patients excluded thus limits generalisability

Abstract is clear re methods and results however I am unsure s to the conclusion – why does the finding suggest that duration of AFT is a prognostic predictor of IPF?

Introduction

Clear and succinct with good use of English language.

Introduction Line 104 – I disagree that “the cause of death is most frequently respiratory failure due to acute exacerbations” AE are variable depending on the studies reviewed. The commonest cause of mortality is respiratory failure due to progressive IPF decline.

In the introduction the authors describe that comorbidities impact quality of life and symptoms. I’m unsure as to the rationale why comorbidities would impact on the efficacy of antifibrotic therapy.

Methods

Ethical approval described.

Please define if patients were diagnosed with an MDT approach or by individual physicians.

Line 142-144 – I’m unsure what this sentence is describing. 5 patients had unfavourable outcome??

How did the physicians make decisions about which antifibrotic to start – I presume this isnt standardised and as a result this would introduce significant bias as comorbidities may influence prescription of each antifibrotic.

Charlson Comorbidity Index score needs to be described in the methods for the readers

Statistics described

Results

66 patients out of 196 consecuative patients were excluded due to various factors including lack of lung function data – this adds significant bias to the study.

Those with low CCIS had higher DLCO -why? And more prescriptions with nintedanib and pirfenidone – implying that patients with low CCIS were more likely to be prescribed antifibrotics

In Table 2

Univariate analysis shows in high CCIS FVC, DLCO and GAP impacted 3 year IPF events which isnt surprising

In low CCIS it was age and duration of AFT – However duration of AFT is biased as there may be lead time bias as patients started AFT at differing stages of their disease

GAP and duration in multivariate analysis

Figures are clear and easy to follow

Tables are clear and easy to follow

Results clearly presented

It would be helpful to know the proportion of patients with each outcome as described eg mortality, AE etc.

Discussion

Authors describe studies that show impact of comorbidities on prognosis in IPF and also that CCIS was a prognostic indicator in IPF – a paper published by this group. This needs to also be added into the introduction to describe the rationale for this study. The authors describe the potential reasons for not seeing a difference in this cohort including this cohort having worse lung function, lower CCSI and previous nintedanib studies show effect despite various sub-groups including CCSI.

Authors conclude that longer duration of AF has impact on prognosis – however the authors haven’t presented data showing that the patients are equally matched re shorter and longer AFT duration eg related to baseline comorbidities, lung function etc which is a limitation of this study – so this conclusion needs to be tempered.

Limitations are discussed re small numbers and generalisability but authors haven’t discussed biases associated with retrospective studies as described above.

6. PLOS authors have the option to publish the peer review history of their article (what does this mean?). If published, this will include your full peer review and any attached files.

Reviewer #1: No

Reviewer #2: **Yes: **Nazia Chaudhuri

---

## [Author Response · Author response to Decision Letter 0]

21 Apr 2023

Reviewer #1

Manuscript ID

PONE-D-22-31881

Manuscript Title

Charlson Comorbidity Index score does not predict the duration and efficacy of antifibrotic treatment for idiopathic pulmonary fibrosis: A multicenter retrospective observational study

Article Type

Research Article

Journal

PLOS ONE

Thank you for taking the time to review our manuscript and for providing your valuable opinions. We have reviewed the text carefully and have answered the comments as follows.

Comment 1

Methods

The authors should explain why a threshold of 3 was chosen for patient selection in low and high CCSI groups. 

【Our reply】

Thank you for very important comment. The enrolled patients were divided into the low CCIS group (≤ 2 points) and the high CCIS group (≥ 3 points), because the average values of CCIS from our data were 2.2 points. We inserted the sentence to the Statistical analysis section (Page 8, line 169).

Comment 2

It is possible that higher value of CCIS could impact on the survival or duration of treatment. I suggest performing a cox regression analysis using CCIS as continuous variable.

【Our reply】

As your suggestion, we performed a cox regression analysis using CCIS as continuous variable and inserted the data as Table 2 (Univariate and multivariate analysis of primary predictors of 3-year IPF related events). GAP score and duration of AFT proved to be the significant predictors.

(Page 13, line 225) Univariate and multivariate analysis were performed to determine the primary predictors of 3-year IPF-related events, using the following parameters: age, sex, CCIS, %DLco, %FVC, GAP score, HRCT UIP pattern, anti-fibrotic agent (nintedanib vs. pirfenidone), and duration of AFT (Table 2). Multivariate analysis showed GAP score and duration of AFT as the significant predictors among all patients.

Comment 3

What AEs were collected and considered as disease outcomes for the purpose of this study? This data is not explained in the methods.

【Our reply】

We have inserted the definition as follows (Page 7, line 141): An AE was defined as significant respiratory deterioration including clinical worsening of dyspnea, hypoxemia, or the worsening or severe impairment of gas exchange characterized by new bilateral ground-glass opacification/consolidation superimposed on a background pattern consistent with IPF pattern not fully explained by cardiac failure or fluid overload [4]. Also, we have inserted the information oabout AE in Table 1.

Comment 4

Results

Add the list of comorbidities and a comparison between the two groups.

【Our reply】

Very important comment. We have inserted the information about the list of comorbidities and compared between the low and high CCIS groups. In the high CCIS group, the incidence of cardiovascular disease and malignancy were higher than those in the low CCIS group.

Comment 5

Add a list of AEs recorded during follow-up and compare the two groups.

【Our reply】

Please see our reply for comment 3. And there was no significant difference for the incidence of AE between the low and high CCIS groups.

Comment 6

Discussion

Authors showed that in low CCIS group, but not in high CCIS group, AFT treatment <1yr impacts on survival. This result deserves to be further explored in the discussion.

【Our reply】

Very important comments. Basically, we think that the timing of initiating AFT and whether to use ninetdanib or pirfenidone depends on the physicians’ decision. Actually, from Table 1, %DLco was preserved in the low CCIS group compared to the high CCIS group, suggesting that AFT was introduced in relatively mild IPF patients in the low CCIS group. Also, as shown in Table 4, we compared the patients’ background between with duration of AFT < 1 year and ≥ 1 year in both low and high CCIS groups and found that there was no significant difference of patients’ characteristics between these groups except %DLco especially in low CCIS group. We newly added the following sentences.

(Page 22, line 316) Consistent with this observation, in the low CCIS group log-rank tests showed significant difference in the Kaplan–Meier survival curves between patients with duration of AFT < 1 year and ≥ 1 year. As shown in Table 4, in the low CCIS group, the patients with duration of AFT ≥ 1 year group (lower %DLco populations than those with duration of AFT < 1 year) had favorable disease outcome, though there was no significant difference in the duration of AFT < 1 year and ≥ 1 year in the GAP score and the time from diagnosis to starting AFT. From these, AFT may be expected to prolong prognosis even in the patients with relatively advanced IPF under appropriate management of comorbidities.

(Page 22, line 329) Second, the timing of initiating AFT and whether to use ninetdanib or pirfenidone depends on the physicians’ decision. Actually, from Table 1, %DLco was preserved in the low CCIS group compared to the high CCIS group, suggesting that AFT was introduced in relatively mild IPF patients in the low CCIS group. In general, for both patient and physician, it is difficult to use anti-fibrotic agents due to their high cost and the difficulty of management of the adverse reaction. From these, it seems that selection bias exists. Furthermore, the different launch dates of pirfenidone (in 2008) and nintedanib (in 2015) in Japan may have contributed to be the limited number of prescriptions, as shown in the supplement figure 2.

 

Reviewer #2

Manuscript ID

PONE-D-22-31881

Manuscript Title

Charlson Comorbidity Index score does not predict the duration and efficacy of antifibrotic treatment for idiopathic pulmonary fibrosis: A multicenter retrospective observational study

Article Type

Research Article

Journal

PLOS ONE

Thank you for taking the time to review our manuscript and for providing your valuable opinions. We have reviewed the text carefully and have answered the comments as follows.

General comment 

This is a retrospective review assessing whether comorbidities predict efficacy of antifibrotics using the Charlson Comorbidity Index score. This is a multi-centre study over a 10 year period.

Only 130 patients are included in this 10 year period – approximately 13 patients per year which is a very low number of patients for a multicentre study – 7 described here. The methods describe that consecutive patients were recruited during this time period. 196 were identified which is approximately 16 IPF patients in 7 centres per month. The study showed no difference in outcome based on whether patients were stratified according to low or high CCIS score.

【Our reply】

Very important comments. As you indicated, this study is a retrospective cohort study with small number IPF patients, and also the initiation of anti-fibrotic agents including pirfenidone and ninetdanib depends on the physicians‘ decision. In addition, it is difficult to use anti-fibrotic agents in all patients with IPF due to their high cost and the difficulty of management of the adverse reaction. From these, it seems that selection bias exists. Furthermore, the different launch dates of pirfenidone (in 2008) and nintedanib (in 2015) in Japan may have contributed to be the limited numbers, as shown in the chart below. Recently, the first choice of antifibrotic drug is nintedanib. I added these contents to the limitation section.

(Page 22, line 326) First, this study is a retrospective cohort study with small number IPF patients and, then the reproducibility of the findings of this study needs to be confirmed through validation cohorts which increased the number of cases in the future. Second, the timing of initiating AFT and whether to use ninetdanib or pirfenidone depends on the physicians’ decision. Actually, from Table 1, %DLco was preserved in the low CCIS group compared to the high CCIS group, suggesting that AFT was introduced in relatively mild IPF patients in the low CCIS group. In general, for both patient and physician, it is difficult to use anti-fibrotic agents due to their high cost and the difficulty of management of the adverse reaction. From these, it seems that selection bias exists. Furthermore, the different launch dates of pirfenidone (in 2008) and nintedanib (in 2015) in Japan may have contributed to be the limited number of prescriptions, as shown in the supplement figure 2.

Comment 2 

Strengths of paper:

This paper is original as I’m not aware that any papers have published on this before. Despite this I’m unsure as to the rationale for why comorbidities would impact on the efficiency of antifibrotic therapy and the authors have failed to explain this rationale in the introduction (although its later described in discussion) Clearly and succinctly written with good use of English language Figures and tables are clear.

【Our reply】

Very important comments. As you indicated, we have failed to explain the rationale for why comorbidities would impact on the efficiency of antifibrotic therapy. Therefore, we added the following to the Introduction and Discussion regarding the impact of comorbidities on the progression of IPF. Also, we added [12] to [15] to the references.

(Page 6, line 117) Although the mechanism of IPF progression is not well known, a previous study has demonstrated that these comorbidities can have a significant influence on the symptoms and quality of life of patients with IPF, particularly when multiple comorbidities are present [11]. Furthermore, in the previous studies, the progression of comorbidities may be pathophysiologically linked to the progression of IPF itself, though their prognostic impact and mechanism is not fully understood [12-15]. The difficulty of management for pirfenidone and nintedanib, the high incidence of older adults, and the various comorbidities seems to affect the tolerability and clinical efficacy of AFT, however, few studies have investigated the clinical impact of AFT in patients with and without comorbidities or evaluated its importance in terms of the long-term prognosis of IPF [16].

(Page 20, line 273) Also, in the recent research the progression of comorbidities reported to be pathophysiologically linked to the progression of IPF itself, though their prognostic impact and mechanism is not fully understood [12-15]. Previous studies have revealed a high incidence of lung cancer in IPF (7% to 20%), though the true cumulative incidence of lung cancer after the diagnosis of IPF and its predictive factors at the initial diagnosis of IPF remain unknown. Various mechanisms such as endoplasmic reticulum stress, alterations of growth factors expression, oxidative stress, and large genetic and epigenetic variations, myofibroblast/mesenchymal transition, myofibroblast activation and proliferation can contributed to predispose the patient to develop IPF and lung cancer [12]. MicroRNAs (miRNAs) regulate gene expression at the post-transcriptional level contributing to all major cellular processes, including oxidative stress and cell death. Several miRNAs have been reported to cross-talk with oxidative stress in both cardiac and pulmonary systems [13]. Fibrogenic mediators such as transforming growth factor-β promotes fibroblast migration, proliferation, and activation in the heart and lungs [14]. Mechanisms contributing to the development of pulmonary hypertension in patients with IPF are complex including hypoxia causing smooth muscle hypertrophy and collagen deposition in pulmonary arteries, the destruction and obstruction of pulmonary vasculature by the progression of pulmonary fibrosis, and vascular remodeling contributed by fibroblast growth factor and platelet-derived growth factor [15]. From these, the progression of IPF seems to cross-talk with other comorbidities, suggesting that comorbidities may have an impact on the efficacy of AFT and high CCIS not only indicates an increased risk of death from comorbidities but also poorer prognosis for IPF itself.

Comment 3

Weaknesses of paper.

Small numbers in 7 centres in Japan limit the generalisability of the results

Limited information in methods on how diagnosis is made and the impact that comorbidities may have on prescribing the antifibrotics or decision making on which antifibrotic is prescribed which is clearly evident in the results. The process isn’t standardised and left to the physicians which introduces a lot of bias. Retrospective review brings in a lot of bias re lead time bias ; diagnostic bias which limits any conclusions from the study and this needs to be tempered in the discussion

A significant number of patients excluded thus limits generalizability.

【Our reply】

I think that's a very valid point. I think that's a very valid point. Since there are many limitations in this research, I added the limitations section as follows.

(Page 22, line 326) First, this study is a retrospective cohort study with small number IPF patients and, then the reproducibility of the findings of this study needs to be confirmed through validation cohorts which increased the number of cases in the future. Second, the timing of initiating AFT and whether to use ninetdanib or pirfenidone depends on the physicians’ decision. Actually, from Table 1, %DLco was preserved in the low CCIS group compared to the high CCIS group, suggesting that AFT was introduced in relatively mild IPF patients in the low CCIS group. In general, for both patient and physician, it is difficult to use anti-fibrotic agents due to their high cost and the difficulty of management of the adverse reaction. From these, it seems that selection bias exists. Furthermore, the different launch dates of pirfenidone (in 2008) and nintedanib (in 2015) in Japan may have contributed to be the limited number of prescriptions, as shown in the supplement figure 2. 

Comment 4

Abstract is clear re methods and results however I am unsure s to the conclusion – why does the finding suggest that duration of AFT is a prognostic predictor of IPF?

【Our reply】

I agree with your point. The sentence that long-term AFT may be an important prognostic indicator in patients with IPF was overstate. We modified the conclusion as follow, “The present results showed that CCIS could not influence the disease outcomes of IPF patients undergoing AFT. (Page 3, line 72)”. Also, we changed the manuscript title as follows, “The clinical impact of comorbidities among patients with idiopathic pulmonary fibrosis undergoing anti-fibrotic treatment: A multicenter retrospective observational study (Page 1, line 1)”.

Comment 5 (Introduction)

Clear and succinct with good use of English language.

Introduction Line 104 – I disagree that “the cause of death is most frequently respiratory failure due to acute exacerbations” AE are variable depending on the studies reviewed. The commonest cause of mortality is respiratory failure due to progressive IPF decline.

In the introduction the authors describe that comorbidities impact quality of life and symptoms. I’m unsure as to the rationale why comorbidities would impact on the efficacy of antifibrotic therapy.

【Our reply】

I agree with these points. We inserted the sentence, “the important causes of death is respiratory failure due to progression of IPF or acute exacerbation (AE) [4]. (Page 6, line 102)”. About the rational why comorbidities would impact on the efficacy of antifibrotic therapy, we replied to the comment 2.

Comment 6 (Methods)

Ethical approval described.

Please define if patients were diagnosed with an MDT approach or by individual physicians.

【Our reply】

We modified the sentence as follows, “The diagnosis of IPF by individual physicians was based on the criteria proposed by the official guideline of the American Thoracic Society/European Respiratory Society/Japanese Respiratory Society/Latin American Thoracic Society [2]. (Page 7, line 139)” Also, we thought it was one of limitations and we added the sentences in the limitation sections as follows, “Third, the diagnosis of IPF depended on individual physicians based on the criteria proposed by official guideline despite the majority of cases having a typical UIP pattern on HRCT. Although this study is useful because it reflects real world data, a multi-disciplinary discussion may be necessary for a rigorous diagnosis of IPF. (Page 23, line 338)”

Line 142-144 – I’m unsure what this sentence is describing. 5 patients had unfavourable outcome??

【Our reply】

Very sorry, this is a simple mistake. We have deleted.

How did the physicians make decisions about which antifibrotic to start – I presume this isn’t standardised and as a result this would introduce significant bias as comorbidities may influence prescription of each antifibrotic.

【Our reply】

Very important comment. We also agree that the timing of initiating AFT and whether to use ninetdanib or pirfenidone depends on the physicians’ decision (Page 22, line 329). Then, we added the information for the baseline data about nintedanib and pirfenidone groups in Supplement Table 1 and found that there was no significant difference of baseline data including age, sex, CCIS, pulmonary function, severity, the timing of starting AFT between nintedanib and pirfenidone groups.

Charlson Comorbidity Index score needs to be described in the methods for the readers

Statistics described

【Our reply】

We added the description for CCIS to the Methods as follows, “The CCIS is an established and widely used tool to measure comorbidity burden with 19 different medical conditions [17]. It is scored based on the number and severity of comorbidities, with higher CCI indicating greater comorbidity burden and severity. It was developed to assess the risk of death from comorbidities and has been widely applied as a prognostic indicator for patients with colorectal cancer, advanced non-small cell lung carcinoma, and acute myocardial infarction [19-21]. (Page 8, line 156)”.

Comment 7 (Results)

66 patients out of 196 consecuative patients were excluded due to various factors including lack of lung function data – this adds significant bias to the study.

【Our reply】

As you indicated, 66 patients out of 196 consecuative patients were excluded due to various factors including lack of lung function data in the present study. We think this adds significant bias to the study. For reference, as shown in supplement figure 1, similar trends were also observed in 169 patients with evaluable CCIS and 3-year IPF-related events. 

Those with low CCIS had higher DLCO -why? And more prescriptions with nintedanib and pirfenidone – implying that patients with low CCIS were more likely to be prescribed antifibrotics

【Our reply】

Very important comment. As you indicated, from Table 1, %DLco was preserved in the low CCIS group compared to the high CCIS group. We speculated that AFT was introduced in relatively mild IPF patients in the low CCIS group (Page 23, line 331). Also, we found that as shown in Table 4, in the low CCIS group, the patients with duration of AFT ≥ 1 year group (lower %DLco populations than those with duration of AFT < 1 year) had favorable disease outcome, though there was no significant difference in the duration of AFT < 1 year and ≥ 1 year in the GAP score and the time from diagnosis to starting AFT. From these, AFT may be expected to prolong prognosis even in the patients with relatively advanced IPF under appropriate management of comorbidities (Page 22, line 316).

In Table 2

Univariate analysis shows in high CCIS FVC, DLCO and GAP impacted 3 year IPF events which isnt surprising. In low CCIS it was age and duration of AFT – However duration of AFT is biased as there may be lead time bias as patients started AFT at differing stages of their disease

GAP and duration in multivariate analysis

【Our reply】

Very important comment. We newly added the information of the patients’ characteristics in the duration of AFT < 1 year and ≥ 1 year using Table 4 and found that in the low CCIS group, the patients with duration of AFT ≥ 1 year group (lower %DLco populations than those with duration of AFT < 1 year) had favorable disease outcome, though there was no significant difference in the duration of AFT < 1 year and ≥ 1 year in the GAP score and the time from diagnosis to starting AFT.

It would be helpful to know the proportion of patients with each outcome as described eg mortality, AE etc.

【Our reply】

We added the information of AE to Table 1 and Table 4. In addition, details of CCIS and adverse events from AFT were added.

Comment 8 (Discussion)

Discussion

Authors describe studies that show impact of comorbidities on prognosis in IPF and also that CCIS was a prognostic indicator in IPF – a paper published by this group. This needs to also be added into the introduction to describe the rationale for this study. The authors describe the potential reasons for not seeing a difference in this cohort including this cohort having worse lung function, lower CCIS and previous nintedanib studies show effect despite various sub-groups including CCIS.

【Our reply】

About the rational why comorbidities would impact on the efficacy of antifibrotic therapy, we replied to the comment 2.

Authors conclude that longer duration of AFT has impact on prognosis – however the authors haven’t presented data showing that the patients are equally matched re shorter and longer AFT duration eg related to baseline comorbidities, lung function etc which is a limitation of this study – so this conclusion needs to be tempered. 

【Our reply】

Very important comment. We newly added the information of the patients’ characteristics in the duration of AFT < 1 year and ≥ 1 year using Table 4 and found that in the low CCIS group, the patients with duration of AFT ≥ 1 year group (lower %DLco populations than those with duration of AFT < 1 year) had favorable disease outcome, though there was no significant difference in the duration of AFT < 1 year and ≥ 1 year in the GAP score and the time from diagnosis to starting AFT. And in the limitation section, we added the following sentences, “Second, the timing of initiating AFT and whether to use ninetdanib or pirfenidone depends on the physicians’ decision. Actually, from Table 1, %DLco was preserved in the low CCIS group compared to the high CCIS group, suggesting that AFT was introduced in relatively mild IPF patients in the low CCIS group. In general, for both patient and physician, it is difficult to use anti-fibrotic agents due to their high cost and the difficulty of management of the adverse reaction. From these, it seems that selection bias exists. Furthermore, the different launch dates of pirfenidone (in 2008) and nintedanib (in 2015) in Japan may have contributed to be the limited number of prescriptions, as shown in the supplement figure 2. (Page 22, line 329)”.

Limitations are discussed re small numbers and generalisability but authors haven’t discussed biases associated with retrospective studies as described above.

Very important comment. Please see the comment 3.

---

## [Decision Letter · Decision Letter 1]

3 Jul 2023

PONE-D-22-31881R1The clinical impact of comorbidities among patients with idiopathic pulmonary fibrosis undergoing anti-fibrotic treatment: A multicenter retrospective observational studyPLOS ONE

Dear Dr. Hara,

Thank you for submitting your manuscript to PLOS ONE. After careful consideration, we feel that it has merit but does not fully meet PLOS ONE’s publication criteria as it currently stands. Therefore, we invite you to submit a revised version of the manuscript that addresses the points raised during the review process.

 The reviewers have recommended publication, but also suggest some minor revisions to your manuscript.  Therefore, I invite you to respond to the reviewers' comments and revise your manuscript.

We look forward to receiving your revised manuscript.

Kind regards,

Fumihiro Yamaguchi

Academic Editor

PLOS ONE

Journal Requirements:

Reviewers' comments:

Reviewer's Responses to Questions

**Comments to the Author**

1. If the authors have adequately addressed your comments raised in a previous round of review and you feel that this manuscript is now acceptable for publication, you may indicate that here to bypass the “Comments to the Author” section, enter your conflict of interest statement in the “Confidential to Editor” section, and submit your "Accept" recommendation.

Reviewer #3: All comments have been addressed

Reviewer #4: (No Response)

Reviewer #5: (No Response)

Reviewer #6: All comments have been addressed

2. Is the manuscript technically sound, and do the data support the conclusions?

Reviewer #3: Partly

Reviewer #4: Partly

Reviewer #5: No

Reviewer #6: Partly

3. Has the statistical analysis been performed appropriately and rigorously? 

Reviewer #3: Yes

Reviewer #4: No

Reviewer #5: I Don't Know

Reviewer #6: I Don't Know

4. Have the authors made all data underlying the findings in their manuscript fully available?

Reviewer #3: Yes

Reviewer #4: No

Reviewer #5: Yes

Reviewer #6: Yes

5. Is the manuscript presented in an intelligible fashion and written in standard English?

Reviewer #3: (No Response)

Reviewer #4: Yes

Reviewer #5: Yes

Reviewer #6: Yes

6. Review Comments to the Author

Reviewer #3: -Please include sample size (sample size calculation) as this can affect the power of the study

-You defined disease out come as " IPF-related mortality due to 142 respiratory failure or the first AE within 3 years after initiating AFT",Not clear how you defined AE, in my opinion, you should have kept death and respiratory failure, if need be you could add SAE or AE that may have led to stoppage or change of medication.

-Is their a reason you used 'opt out' instead of 'opt-in' in the consent?

-Remove information in the 'conclusion/discussion ' that you have not provided data in the text

-'The long period of AFT is a predictor of outcome'- is this a factor of long AFT or the disease period?

Reviewer #4: This retrospective study has tried to find out whether Charlson Comorbidity Index Score (CCIS) can predict efficacy of anti-fibrotic treatment (AFT) for IPF patients. There are several limitations to this study which has been pointed out by previous reviewers and has been graciously recognized by the authors as well.

1) The sample size is small for a retrospective study spanning 10 years and 7 centers

2) The choice of treatment with nintedanib and pirfenidone was not uniform and there was likely selection bias for the same.

3) There was lead time bias, indicated by the fact that in low CCIS group, there was AFT initiation in relatively early part of IPF disease course (Shown by relatively preserved DLCO for the same).

Additional Comments:

4) In the statistical analysis part, it should be mentioned how checking of the data for normal or non-normal distribution was done. (Ex: Using tests like Shapiro-Wilk or Kolmogorov-Smirnov etc..). This would point to and strengthen the rationale for using mean and SD or median and IQR for continuous variables.

5) It is not clear whether the distribution of CCIS scores overall was normally distributed or not. If not, then the criteria for splitting into low and high score groups should be done by their median and not by their mean value. (Lines 172-174)

6) The study has failed to show any difference in outcomes between low and high CCIS groups of IPF patients. However, absence of proof should not be taken as proof of absence. Moreover, even if there had been a difference, it would have been an association and not necessarily a causation. So the conclusion must be carefully written, something like, "In our study, we could not show any relation between CCIS scores and IPF disease outcomes in patients undergoing treatment with anti-fibrotics."

7) Finally, it was not mentioned whether any patients on nintedanib or pirfenidone faced any complications which prompted a change in dosing or a switch over to the other drug. The pill burden also is usually more for pirfenidone than nintedanib. A lot of patients who suffer from GI complications with pirfenidone or are not comfortable with the pill burden have to be switched over to nintedanib. Such a change however will cause more bias introduced by the treatment giver in the results.

Reviewer #5: Dear Authors

You responded to all question very well.

But I am not sure that the question of research is appropriate.

Until know the benefit of antifibrotic agent on pulmonary fibrosis is modest. So discussion about effect of antifibrotic agent on comorbidities is not make sense.

Reviewer #6: It is interesting to focus on the cinical impact of comorbidities on antifibrotic treatment specifically among these patients.

7. PLOS authors have the option to publish the peer review history of their article (what does this mean?). If published, this will include your full peer review and any attached files.

Reviewer #3: No

Reviewer #4: **Yes: **Dr. Saikat Banerjee, MD Respiratory Medicine, MIT Micromasters in Statistics and Data Science

Reviewer #5: No

Reviewer #6: **Yes: **Khadija Ayed

---

## [Author Response · Author response to Decision Letter 1]

6 Aug 2023

Reviewer #3

Manuscript ID

PONE-D-22-31881R1

Manuscript Title

The clinical impact of comorbidities among patients with idiopathic pulmonary fibrosis undergoing anti-fibrotic treatment: A multicenter retrospective observational study

Article Type

Research Article

Journal

PLOS ONE

Thank you for taking the time to review our manuscript and for providing your valuable opinions. 

We have reviewed the text carefully and have answered the comments as follows.

Comment 1

Please include sample size (sample size calculation) as this can affect the power of the study.

Our reply

Very important comment, however, no statistical sample size calculations were conducted because of the retrospective nature. So, we inserted the sentences into “Statistical analysis” and “limitation” sections.

Comment 2

You defined disease out come as " IPF-related mortality due to 142 respiratory failure or the first AE within 3 years after initiating AFT", Not clear how you defined AE, in my opinion, you should have kept death and respiratory failure, if need be you could add SAE or AE that may have led to stoppage or change of medication.

Our reply

Thank you for the important comment. In the manuscript, the definition of AE was unclear. So, we re-defined AE as acute respiratory deterioration requiring steroid pulse therapy adding on AFT including clinical worsening of dyspnea, hypoxemia, or the worsening or severe impairment of gas exchange characterized by new bilateral ground-glass opacification/consolidation superimposed on a background pattern consistent with IPF pattern not fully explained by cardiac failure or fluid overload [4]. From the randomized control trial evaluating pirfenidone and nintedanib, the event of AE was very important endpoint [8, 9] and we define as the endpoint as 3-year IPF-related events including IPF-related mortality due to respiratory failure or the first AE within 3 years after initiating AFT. Also, we evaluated the survival curves for 3-year IPF-related mortality according to CCIS in Supplement Figure 2 and demonstrated that comparison of the curves by log-rank test according to CCIS group (A), and according to CCIS group in patients treated with nintedanib (B) and pirfenidone (C) showed no significant difference (P = 0.30, P = 0.21, and P = 0.91, respectively).

Comment 3

Is there a reason you used 'opt out' instead of 'opt-in' in the consent?

Our reply

In this study, there are many death cases, it may be difficult to obtain direct consent from blood relatives, and we would like to analyze all cases that underwent AFT as much as possible. After discussion with the IRB at each institution, opt-in was used.

Comment 4

Remove information in the 'conclusion/discussion ' that you have not provided data in the text.

Our reply

We remove the sentence “(CCIS of GAP stage I, 2 points; II, 2 points; III, 2.5 points (P = 0.32))” and “Furthermore, the different launch dates of pirfenidone (in 2008) and nintedanib (in 2015) in Japan may have contributed to be the limited number of prescriptions, as shown in the supplement figure 2.” in the discussion and moved to results section.

Comment 5

'The long period of AFT is a predictor of outcome'- is this a factor of long AFT or the disease period?

Our reply

Sorry, the sentence was confused. We inserted the sentence, “Especially, in the low CCIS group but not the high CCIS group, the longer duration of AFT had better disease outcome.” In the Abstract section.

 

Reviewer #4

Manuscript ID

PONE-D-22-31881R1

Manuscript Title

The clinical impact of comorbidities among patients with idiopathic pulmonary fibrosis undergoing anti-fibrotic treatment: A multicenter retrospective observational study

Article Type

Research Article

Journal

PLOS ONE

Thank you for taking the time to review our manuscript and for providing your valuable opinions. 

We have reviewed the text carefully and have answered the comments as follows.

Comment 1

In the statistical analysis part, it should be mentioned how checking of the data for normal or non-normal distribution was done. (Ex: Using tests like Shapiro-Wilk or Kolmogorov-Smirnov etc..). This would point to and strengthen the rationale for using mean and SD or median and IQR for continuous variables.

Comment 2

It is not clear whether the distribution of CCIS scores overall was normally distributed or not. If not, then the criteria for splitting into low and high score groups should be done by their median and not by their mean value. (Lines 172-174).

Our reply

These comments were very important. We inserted the sentence in the Statistical Analysis section as follows.

―Groups were compared using chi-square test and Wilcoxon rank-sum test (non-parametric test) because a normal distribution cannot be assumed for data obtained with a small number of samples.

―The enrolled patients were divided into the low CCIS group (≤ 2 points) and the high CCIS group (≥ 3 points), because the average and median values of CCIS from our data were 2.2 and 2 points, respectively.

Comment 3

The study has failed to show any difference in outcomes between low and high CCIS groups of IPF patients. However, absence of proof should not be taken as proof of absence. Moreover, even if there had been a difference, it would have been an association and not necessarily a causation. So the conclusion must be carefully written, something like, "In our study, we could not show any relation between CCIS scores and IPF disease outcomes in patients undergoing treatment with anti-fibrotics."

Our reply

Very important comment. We inserted the sentence to the conclusion section as you indicated as follow. 

“In the present study, we could not show any relation between CCIS and IPF disease outcomes in patients undergoing AFT, though the longer duration of AFT might be beneficial for IPF outcomes among patients with low CCIS.”

Comment 4

Finally, it was not mentioned whether any patients on nintedanib or pirfenidone faced any complications which prompted a change in dosing or a switch over to the other drug. The pill burden also is usually more for pirfenidone than nintedanib. A lot of patients who suffer from GI complications with pirfenidone or are not comfortable with the pill burden have to be switched over to nintedanib. Such a change however will cause more bias introduced by the treatment giver in the results.

Our reply

Very important comment. We added the information of the number and proportion of AFT dose reduction and discontinuation patients. And in the present study, there was no switch between nintedanib and pirfenidone.

 

Reviewer #5

Manuscript ID

PONE-D-22-31881R1

Manuscript Title

The clinical impact of comorbidities among patients with idiopathic pulmonary fibrosis undergoing anti-fibrotic treatment: A multicenter retrospective observational study

Article Type

Research Article

Journal

PLOS ONE

Thank you for taking the time to review our manuscript and for providing your valuable opinions. 

We have reviewed the text carefully and have answered the comments as follows.

Comment 1

You responded to all question very well. But I am not sure that the question of research is appropriate. Until know the benefit of antifibrotic agent on pulmonary fibrosis is modest. So discussion about effect of antifibrotic agent on comorbidities is not make sense.

Our reply

Thank you for your comment. I agree with your opinion which the benefit of antifibrotic agent on pulmonary fibrosis is modest. We think that the progression of IPF seems to cross-talk with other comorbidities and the longer duration of AFT might be beneficial for IPF outcomes among patients with low CCIS from our data. Therefore, in the present study, we would like to emphasize the importance of evaluating comorbidities in patients with IPF and the optimal use of AFT accordingly may improve the long-term prognosis of IPF.

---

## [Decision Letter · Decision Letter 2]

31 Aug 2023

The clinical impact of comorbidities among patients with idiopathic pulmonary fibrosis undergoing anti-fibrotic treatment: A multicenter retrospective observational study

PONE-D-22-31881R2

Dear Dr. Hara,

We’re pleased to inform you that your manuscript has been judged scientifically suitable for publication and will be formally accepted for publication once it meets all outstanding technical requirements.

Kind regards,

Fumihiro Yamaguchi

Academic Editor

PLOS ONE

Additional Editor Comments (optional):

Reviewers' comments:

Reviewer's Responses to Questions

**Comments to the Author**

1. If the authors have adequately addressed your comments raised in a previous round of review and you feel that this manuscript is now acceptable for publication, you may indicate that here to bypass the “Comments to the Author” section, enter your conflict of interest statement in the “Confidential to Editor” section, and submit your "Accept" recommendation.

Reviewer #4: All comments have been addressed

2. Is the manuscript technically sound, and do the data support the conclusions?

Reviewer #4: (No Response)

3. Has the statistical analysis been performed appropriately and rigorously? 

Reviewer #4: (No Response)

4. Have the authors made all data underlying the findings in their manuscript fully available?

Reviewer #4: (No Response)

5. Is the manuscript presented in an intelligible fashion and written in standard English?

Reviewer #4: (No Response)

6. Review Comments to the Author

Reviewer #4: (No Response)

7. PLOS authors have the option to publish the peer review history of their article (what does this mean?). If published, this will include your full peer review and any attached files.

Reviewer #4: **Yes: **Dr. Saikat Banerjee, MD Respiratory Medicine, Micromasters (MIT) in Statistics and Data Science

---

## [Editor Report · Acceptance letter]

8 Sep 2023

PONE-D-22-31881R2 

The clinical impact of comorbidities among patients with idiopathic pulmonary fibrosis undergoing anti-fibrotic treatment: A multicenter retrospective observational study 

Dear Dr. Hara:

I'm pleased to inform you that your manuscript has been deemed suitable for publication in PLOS ONE. Congratulations! Your manuscript is now with our production department. 

Kind regards, 

on behalf of

Dr. Fumihiro Yamaguchi 

Academic Editor

PLOS ONE